# Transcriptional Profiling and Biological Pathway(s) Analysis of Type 2 Diabetes Mellitus in a Pakistani Population [note 1]

**DOI:** 10.3390/ijerph17165866

**Published:** 2020-08-13

**Authors:** Zarish Noreen, Christopher A. Loffredo, Attya Bhatti, Jyothirmai J. Simhadri, Gail Nunlee-Bland, Thomas Nnanabu, Peter John, Jahangir S. Khan, Somiranjan Ghosh

**Affiliations:** 1 HealthCare Biotechnology, Atta-Ur-Rahman School of Applied Biosciences, National University of Sciences and Technology (NUST), Islamabad 44000, Pakistan; zarish.noreen@gmail.com (Z.N.); attyabhatti@gmail.com (A.B.); pjohn@asab.nust.edu.pk (P.J.); 2Departments of Oncology and of Biostatistics, Georgetown University, Washington, DC 20057, USA; cal9@georgetown.edu; 3Department of Pediatrics and Child Health, College of Medicine, Howard University, Washington, DC 20059, USA; jyothirmai.simhadri@howard.edu (J.J.S.); gnunlee-bland@howard.edu (G.N.-B.); 4Department of Biology, Howard University, Washington, DC 20059, USA; tnnanabu@gmail.com; 5Department of Surgery, Rawalpindi Medical College, Rawalpindi, Punjab 46000, Pakistan; jskdr@hotmail.com

**Keywords:** type 2 diabetes, Pakistan, gene expression, disease pathways, gene validation, biomarkers

## Abstract

The epidemic of type 2 diabetes mellitus (T2DM) is an important global health concern. Our earlier epidemiological investigation in Pakistan prompted us to conduct a molecular investigation to decipher the differential genetic pathways of this health condition in relation to non-diabetic controls. Our microarray studies of global gene expression were conducted on the Affymetrix platform using Human Genome U133 Plus 2.0 Array along with Ingenuity Pathway Analysis (*IPA*) to associate the affected genes with their canonical pathways. High-throughput qRT-PCR TaqMan Low Density Array (TLDA) was performed to validate the selected differentially expressed genes of our interest, viz., *ARNT, LEPR, MYC, RRAD, CYP2D6*, TP53, *APOC1, APOC2, CYP1B1, SLC2A13,* and *SLC33A1* using a small population validation sample (n = 15 cases and their corresponding matched controls). Overall, our small pilot study revealed a discrete gene expression profile in cases compared to controls. The disease pathways included: *Insulin Receptor Signaling*, *Type II Diabetes Mellitus Signaling, Apoptosis Signaling*, *Aryl Hydrocarbon Receptor Signaling*, *p53 Signaling*, *Mitochondrial Dysfunction, Chronic Myeloid Leukemia Signaling*, *Parkinson’s Signaling, Molecular Mechanism of Cancer,* and *Cell Cycle G1/S Checkpoint Regulation, GABA Receptor Signaling, Neuroinflammation Signaling Pathway, Dopamine Receptor Signaling, Sirtuin Signaling Pathway*, *Oxidative Phosphorylation*, *LXR/RXR Activation*, and *Mitochondrial Dysfunction,* strongly consistent with the evidence from epidemiological studies. These gene fingerprints could lead to the development of biomarkers for the identification of subgroups at high risk for future disease well ahead of time, before the actual disease becomes visible.

## 1. Introduction

Type 2 Diabetes Mellitus (T2DM), the predominant form of diabetes, is a chronic, multifactorial, metabolic disorder characterized by hyperglycemia and has become a public health challenge worldwide [1,2,3]. T2DM accounts for 90% to 95% of all diabetes cases with the heritability ranging between 40% and 70% [4,5]. Beta cell dysfunction, impaired insulin secretion, and increased insulin resistance cause the chronic dysregulation of the hyperglycemia state and are critical in pathophysiological determinants of the disease’s progression and pathogenesis [6,7]. T2DM can lead to life-threatening complications, viz., neuropathy, cerebrovascular disease, nephropathy, peripheral vascular disease, coronary artery disease, and retinopathy [8,9]. 

T2DM has emerged as a global pandemic affecting 451 million people worldwide [10]. Globally, the number of affected people has quadrupled in the past three decades, and diabetes mellitus is the ninth major cause of death. Asia is a major area of the rapidly emerging T2DM global epidemic, with China and India the top two epicenters [11]. Pakistan is close behind [12]. The occurrence of T2DM varies extensively from one geographical area to another and within nations as well, as is typical for a multifactorial disorder. Complex interactions between genetic factors and environmental risk factors contribute to T2DM’s pathogenesis [13,14,15]. The environmental risk factors like sedentary lifestyle, physical inactivity [16], high caloric intake, obesity, high consumption of red meat [17], alcohol consumption, and smoking have been recognized as significant contributors [18]. 

According to a Centre for Disease Control and Prevention (CDC) report, 34.2 (10.5%) million people in the USA have diabetes in 2020, with 8 million adults aged 18 years or older having prediabetes (34.5% of the adult US population) [19]: (https://www.cdc.gov/diabetes/data/statistics/statistics-report.html). The worldwide diabetes prevalence for 2019 is estimated to be 9.3% with 463 million people and is expected to rise 10.2% by 2030 with 578 million people and 10.9% by 2045 including 700 million people. It has been observed that the worldwide prevalence of diabetes is higher in urban areas (10.8%) as compared to rural areas (7.2%) and in high-income countries (10.4%) as compared to low income countries (4.0%) [20]. The prevalence of T2DM in all Asian countries is rising and is expected to rise further in future, imposing huge economic burdens on their healthcare systems [21,22]. In South Asia, India is deemed as the world’s capital of diabetes. The diabetic population in the country is close to hitting the alarming mark of 69.9 million by 2025 and 80 million by 2030, an increase of 266% [23]. The World Health Organization (WHO) has projected that T2DM is expected to be the seventh primary cause of death by 2030, when low- and middle-income countries will account for more than 80% of all deaths related to diabetes [24]. 

Research has revealed that in the last two decades many developing countries have been embracing aspects of the Western lifestyle including over consumption of high caloric food and lack of physical activity. About 90% of T2DM subjects are exclusively attributed to excess weight [25]. The prevalence of T2DM and prediabetes is much higher than previously thought in Pakistan [26]. The T2DM prevalence is increasing dramatically in Pakistan as compared to other countries due to growing urbanization and adopting western lifestyle, making it the sixth highest country for T2DM prevalence in the world [27,28]. According to the International Diabetes Foundation (IDF), it is estimated that almost 11.5 million people in Pakistan will have diabetes by the year 2025, which will place Pakistan fifth by the IDF ranking [29,30]. The current prevalence of T2DM in Pakistan is 11.77% overall, slightly higher in males (11.20%) than females (9.19%), and there is additional variation across regions of the country, from <9% to 16% [31]. 

Many people are exposed to polluted environments and increasing evidence has shown epidemiological associations between exposures to persistent organic pollutants (POPs) and the risk of T2DM [32]. A study of selected POPs in the NHANES 1999–2002 data set has reported an association between waist circumferences and Body Mass Index (BMI) in subjects with detectable levels of POPs, perhaps suggesting that these chemicals are plausible contributors to the obesity epidemic [33]. Supporting that hypothesis, a prospective analysis in the Nurses’ Health Study based on two prospective studies and Meta-analysis indicates that higher plasma Hexachlorobenzene (HCB) and total Polychlorinated Biphenyls (PCB) concentrations at baseline were significantly associated with increased incidence of diabetes development [34]. In the Nurses’ Health Study II—a nested case-controlled study involving middle-aged US women—conducted on plasma-POPs (including 3 organochlorine pesticides [OCPs] and 20 polychlorinated biphenyls [PCBs]) indicated a higher risk of T2DM during more than 11 years of follow up. Age, breastfeeding history, previous weight change, and concurrent body weight were among the primary determinants of circulating POP concentrations [35]. 

Our team recently published the results of a descriptive study of Pakistani patients with T2DM, which confirmed gender differences and suggested possible associations with environmental factors [1]. The present study extends those observations to investigate a pilot study of global gene expression patterns in the same patients, using microarrays coupled with Ingenuity Pathway Analysis (IPA) and followed by selective gene validation. The results have informed our insights into the underlying gene expressions changes that may pose an early sign of developing comorbidities in later life.

## 2. Methods

### 2.1. Study Participants, Ethics, and Selection of Participants for Gene-Expression Assays

We randomly selected our study subjects from our previously reported cohort recruited from the out-patient departments of four different regional hospitals of Pakistan, i.e., Islamabad, Rawalpindi, Lahore, and Abbottabad (see details in [1]). The study was undertaken with the prior approval by the Howard University Institutional Review Board (IRB-17-MED-43), and the informed consent was obtained from volunteers as per the approval of the Institutional Review Board (IRB) of Atta-ur-Rahman School of Applied Biosciences (ASAB), National University of Sciences and Technology (NUST) (28/IRB; dated. 20 April 2016).

We have retrieved the detailed background information of our participants from our earlier cohort report [1], where besides age and gender, their HbA1C, plasma glucose (mg/dl), BMI, locality, literacy, history of T2DM, medication use, and smoking history were also captured (see additional information in Appendix A). This review of the records established that the study subjects in the present report were free of obesity (high BMI) and had abstained from use of alcohol (due to cultural practice) The subjects were 2 adults from the T2DM group and 2 from the control group (one male and female from each group): they were gender- and weight- matched, with average age of 68 ± 14.1 years. HbA1C (13.35 ± 0.15%), fasting plasma glucose level (315 ± 35 mg/dl), and BMI (29 ± 0.5) in our T2DM subjects were compared to controls (HbA1: 4.9 ± 0.1 %), fasting plasma glucose level: 113 ± 15 mg/dl, and BMI: 21.5 ± 1.3), and the differences were in the expected directions. Smokers were excluded from selection. Other than T2DM, the selected subjects were free of other chronic diseases and had not undergone any major surgical procedures that required general anesthesia during the last 5 years.

### 2.2. Blood Collection and RNA Preparation

Whole blood was collected in PAXgene^®^ Blood RNA Tubes (BD, Cat. # 762165) in Pakistan from the outpatient department of the hospitals by experienced phlebotomist and were dispatched later through a specialized courier service under frozen conditions (dry ice) to the Department of Biology at Howard University, Washington DC, USA, according to the specific protocol of the manufacturer (Qiagen MD). RNA was extracted from the PAXgene tubes using the PAXgene Blood RNA Kit IVD (Qiagen, Cat. # 762164 of PreAnalytiX GmbH, Germany), according to the manufacturer’s instructions. Contaminating DNA was removed with the Invitrogen DNA-free kit (ThermoFisher, CA, Cat # AM 1906). RNA content was determined spectrophotometrically on a nanodrop at 230, 260 and 280 λ. RNA quality was also verified by Agilent bioanalyzer analysis using an RNA 6000 nanochip before microarray chip hybridization, with RNA stored at −80 °C.

### 2.3. cDNA Synthesis and Microarrays

Total RNA was reverse-transcribed to cDNA by using High-Capacity cDNA Reverse Transcription Kits (Part # 4387406; Applied Biosystems, CA, USA) according to manufacturer’s instruction. The reaction mixture (20 μL total volumes) was incubated at 25 °C for 10 min and then at 37 °C for 60 min followed by 95 °C for 5 min. Finally, the mixture was heated at 95 °C for 5 min. The cDNAs were stored in −15 to −25 °C, if not used immediately (within 24 h), or stored in 2 to 8 °C (See [36,37,38] for detailed methods). 

The oligonucleotide microarray experiments were conducted by EpigenDx (Boston, MA, USA) using the Affymetrix U133 Plus 2.0 Array platform, which has a comprehensive coverage of the whole transcribed human genome on a single array. Chips were run in triplicate form on each individual subject. Expression profiling data were obtained for each sample individually with standard operating procedures and following the protocols of [39] for quality control (also see detailed procedure in [37,40]). 

### 2.4. Gene Expression Data Analysis

Differentially expressed gene sets were analyzed with microarray results using a one-way ANOVA model by Partek Genomics Suite (Partek GS). Briefly, the raw microarray values from the Affymetrix.cel files were imported into Partek Genomics Suite, (Partek Inc., St. Louis, MI, USA). The probe summarization and probe set normalization were done using the GC-RMA algorithm [41], which included GC-RMA background correction, quantile normalization, log2 transformation and median polish probe set summarization. The results yielded the expected proportion of truly null hypotheses among all the rejected null hypotheses [42] and kept the false positive rate below five percent. We selected gene sets that had the statistical significance level < 0.05. This analysis generated two highly significant sets of differentially expressed genes that originated from comparing the T2DM dataset with Control subjects (see details in [38,43]).

### 2.5. Identification of Cellular Processes and Pathways Involved by Ingenuity Pathways Analysis (IPA^®^)

Data sets containing gene identifiers and corresponding expression values (fold change) from the microarray experiment were uploaded into Ingenuity Pathway Analysis software (Ingenuity^®^ Systems, www.ingenuity.com). Each gene identifier was plotted to its similar gene entity in the Ingenuity Pathways Knowledge Base. To advance our pathway analysis, we incorporated an additional 527 gene transcripts from the IPA knowledge base to our results. We used the data sources from Ingenuity expert findings and used the “Core Analysis” function to interpret the data in the perspective of biological processes, pathways, and networks. Differentially expressed gene identifiers were defined as value parameters for analysis and identified the relationship between gene expression alterations and related changes in biofunctions under the subcategories of Molecular and Cellular Functions, Physiological System Development and Function, and Disease and Disorders. Genes differentially expressed with *p* < 0.05 were overlaid onto global molecular networks developed from information contained in the IPA knowledge base. Networks were then algorithmically generated based on their connectivity. Networks were “named” on the most prevalent functional group(s) present. Canonical Pathway (CP) analysis identified function specific genes significantly present within the networks.

### 2.6. Validation by High-Throughput TaqMan^®^ Low Density Array (TLDA)

To validate the altered gene expressions that emerged from the microarray experiment, we used predesigned TLDA cards (in a 16 Format, Applied Biosystems^®^, CA, USA) to examine the expression of genes of interest, viz., *18s* (*Hs99999901_s1;* manufacturing control), *GAPDH* (*Hs99999905_m1;* Internal control)*, ACTBL2, APC, ARNT, ATP1B1, BCL2, CCK, CD3G, CYP2D6, ENTPD3, LEPR, LRP12, MYC, RRAD, APOC1, APOC2, CYP1B1, SLC2A13, SLC33A1, TP53, MYC,* as identified in the methods described above. We randomly selected an additional 15 cases (see Appendix A) and their corresponding controls, gender-, age-, and weight-matched, from the original cohort for TLDA analysis. The details of the experimental procedures are reported in [37,40]. 

### 2.7. TLDA Data Analysis

The TLDA data were analyzed by SDS Ver. 2.4 software (ABI, CA). Threshold cycle (Ct) data for all target genes and the internal control gene 18s RNA were used to calculate ^Δ^Ct values [^Δ^Ct = Ct (target gene) − Ct (18s RNA)]. Then, ^ΔΔ^Ct values were calculated by subtracting the calibrator (control) from the ^Δ^Ct values of each target. To visualize and further expression analysis, the data were exported in plate centric format to RQ Manager (ABI, V 2.4), which allowed us to inspect the status of each gene in the study population. 

## 3. Results

### 3.1. Differential Expression of Genes of T2DM Subjects

Through our microarray gene expression studies, a total number of 31,136 genes were observed to have differential expression in the subjects analyzed (among a total of 53,618 total transcripts on the gene chip). Of these, a total of 1657 genes were significantly different between cases and controls (*p* value < 0.05), among which 61% (1017 genes) were downregulated, and 39% (640 genes) were upregulated with a fold change ranging between −189 (downregulation) to +14 (upregulation) (Appendix A). These significantly expressed genes were used for downstream IPA analysis, as shown below.

### 3.2. The Attribution of Differentially Expressed Genes to Their Biofunctions and Associated Diseases and Disorders

The list of potential biological effects due to differential gene expression in our pilot study can be described in three different levels: canonical pathways and biofunctions (Table 1), and networks (Table 2). Analysis of the genes identified above revealed 5 significant genetic networks (score ≥ 8.0), as listed in Table 2. The top-scoring networks reflect associations with diseases and disorders that primarily include: *Endocrine System Disorders*, *Genetic Disorders*, *Respiratory Disease*, *Cancer*, *Dermatological Disease and Conditions*, *Developmental Disorders*, *Hematological Disease*, *Immunological Disease*, *Connective Tissue Disorder, Hereditary Disorder,* and *Skeletal* and *Muscular Disorders* (Figure 1; panel A). In the Physiological System Development and Functions network, notable functions were *Nervous System Development and Function*, *Tissue Morphology, Tissue Development*, *Hematological System Development and Function*, *Immune Cell T Trafficking*, *Endocrine System Development and Function, Connective Tissue Development and Function*, and *Visual System Development and Function* (Figure 1, panel B). Among the genes in the Molecular and Cellular Functions network, *Lipid Metabolism, Cell Cycle, Cellular Assembly and Organization*, *Cellular Functions and Maintenance*, *Small Molecule Biochemistry*, *Cell Death and Survival*, *Cellular Development*, *Cellular Growth and Proliferation*, and *Cell-To-Cell Signaling and Interaction functions* were revealed (Figure 1; panel C). 

### 3.3. The Canonical Pathways (CP) and Gene Ontology (GO) Enrichment of Biological Processes

In the canonical pathway analysis, we chose to build the pathways connecting the top 3 networks (Networks 1–3, Total Score ≥ 30, Table 2). The top eleven (11) pathways identified through this approach were: *Insulin Receptor Signaling*, *Type II Diabetes Mellitus Signaling, Apoptosis Signaling*, *Aryl Hydrocarbon Receptor Signaling*, *p53 Signaling*, *Mitochondrial Dysfunction, Chronic Myeloid Leukemia Signaling*, *Parkinson’s Signaling, Molecular Mechanism of Cancer, Glioma Signaling,* and *Cell Cycle G1/S Checkpoint Regulation* (Figure 2). Further in-depth analysis also identified some additional pathways, i.e., *GABA Receptor Signaling, Neuroinflammation Signaling Pathway, Dopamine Receptor Signaling, Sirtuin Signaling Pathway*, *Oxidative Phosphorylation*, *LXR/RXR Activation*, and *Mitochondrial Dysfunction* (Figure 3). 

### 3.4. Validation of Selected Genes through TLDA 

Our validation set of cases and controls revealed that the genes *ARNT, LEPR, RRAD,* and *CYP2D6* were strongly upregulated in T2DM subjects compared to controls, except in a few individuals (Table 3; Figure 4). In addition, *ARNT*, *LEPR*, and *RRAD* genes were mostly upregulated in T2DM patients below 70 years of age, while in the older age group *ARNT, LEPR,* and *RRAD* were either upregulated or downregulated (data not shown). The results also showed that *TP53* and *MYC* genes were mostly downregulated in our studied subjects (Figure 4), while expression of *APOC1, APOC2, CYP1B1, SLC2AB,* and *SLC33A1* genes was highly upregulated in all subjects (Figure 5). IPA core analysis of the 29 signaling pathways we report here also revealed that the majority were downregulated (Figure 6), which corroborated our validation analysis.

## 4. Discussion

The delayed diagnosis of T2DM and its insidious inception are the main causes of the large number of mortality and morbidity cases worldwide, especially in developing countries like Pakistan [44,45]. The current pilot study, the first ever reported from that country, examined the gene expression profile data, significant signaling pathways, and distinct biological mechanisms associated with T2DM by high-throughput microarray analysis coupled with IPA analysis. Our gene expression results, however, indicated “cancer” genes as being differentially expressed in T2DM cases compared to controls, which was revealed through the IPA analysis (Figure 1), network connectivity (Figure 2, where *TP53* stands as a central molecule in *P53 Signaling Pathway*), as well as a small pilot population validation of *TP53* gene (Figure 4), where this tumor suppression gene is mostly downregulated. Combining all the observed results, there is a strong suggestion that continued observation and surveillance of this cohort is warranted in regard to potential cancer risk. The resulting differential gene expression patterns probably reflect the common effects of several underlying biological functions, wherein a single gene can be involved in multiple biological processes. Thus, it is imperative to understand gene regulatory pathways and networks that control phenotypically relevant biological mechanisms and extract the most biologically relevant biomarkers to assess the current disease state and to anticipate the possible future comorbid risks in this population.

### 4.1. Genes Associated with T2DM

In the current study, we identified several candidate genes (*ARNT, LEPR, RRAD, CYP2D6, APOC1, APOC2, CYP181, SLC2A13, SLC33A1, MYC,* and *TP53)* which presented statistically significant gene expression profiles. Here, we briefly summarize the knowledge base for each of these genes, as reported in the literature, and speculate on its potential role in T2DM.

*ARNT* (Aryl Hydrocarbon Receptor Nuclear Translocator), also known as *HIF1β* (hypoxia inducible factor-1β), is a heterodimer of the basic helix-loop-helix (bHLH-PAS) family of transcription factors including *HIF1α, HIF2α,* and *AhR. ARNT* is positioned on human chromosome 1q21–q24, and research has revealed that this region has presented a well replicated linkage to T2DM; alteration in the *ARNT* gene increases the risk of T2DM by impaired insulin secretion [46]. *ARNT* forms heterodimeric complexes with *HIF1α, HIF2α*, and *AhR* and is responsible for regulating the gene expression involved in glucose metabolism and glucose transport [47]. ARNT is required to stimulate beta-cells to increase insulin production, and expression of the transcription factor of ARNT is reduced in the islets of humans with T2DM [48]. In our pilot study, the ARNT gene expression was upregulated (Figure 4,) and may have also triggered the down regulation of *FOX01* and *AKT* with *Insulin Receptor Signaling* and *Type 2 Diabetes Mellitus Signaling* among the important canonical pathways (Figure 2), consistent with prior reports [49].

Two other genes of interest in our diabetic subjects, viz., *LEPR* and *RRAD*, are marker proteins for metabolic dysfunction, which were mostly upregulated in our validation study (Figure 4). Leptin, a precursor of adipokines, is secreted by adipose tissues and acts by stimulating leptin receptor (*LEPR*) once it reaches the central nervous and peripheral nervous systems through the circulation. *LEPR* gene is located on chromosome 1p31 and is involved in the blood glucose, insulin secretion, and lipid metabolism [50,51]. *LEPR* gene is present in pancreatic beta cells and is expressively associated with chronic hyperglycemia, uncontrolled T2DM, and diabetes related disorders [52,53,54,55]. In our validation experiment, the *LEPR* gene was upregulated in 93% of the cases. This observation is concordant with previous studies, which showed upregulated gene expression of *LEPR* in this disease [56,57,58,59]. It was also identified as a future disease risk marker for metabolic dysfunction in a Slovakian population with severe environmental chemical exposures [36]. The strong associations between *LEPR* genetic variations and mutations and risk of T2DM has been reported by population-based studies [60,61,62,63].

*RRAD* is a member of the Ras-related family of small GTP binding proteins. Rad 35-kDa (Ras Associated with Diabetes) Ras-guanosine triphospha-tase (GTPase) is positively expressed in response to insulin [64]. The over expression of *RRAD* inhibits the glucose uptake and is associated with insulin resistance and T2DM [65]. We observed that *RRAD* gene expression was mostly upregulated in the cases we studied (80%, Table 3, Figure 4). This is in accord with the other studies that demonstrated increased expression level after giving insulin therapy or treatment [66]. *RRAD’s* primary target of expression is skeletal muscle, which is considered as a major site for glucose disposal and insulin resistance. Studies have also demonstrated its overexpression in the muscles of diabetic patients compared to controls [67,68].

Cytochrome P450 (CYP) is a large family of integral membrane proteins and its CYP isoenzymes involve almost 57 CYP genes in human [69]. These proteins have a heme group and are involved in the oxidative metabolism of many endogenous and exogenous compounds. *CYP2D6* and *CYP1B1,* which we found to be upregulated in diabetic subjects, are two important CYP genes, which are functionally polymorphic and mediate an estimated 40% of all drug metabolism [70]. *CYP1B1* catalyzes xenobiotic metabolism and is regulated by transcription factors like AHR nuclear translocator (*ARNT*) complex (*AHR/ARNT*) [71,72]. Our results agree with previous studies in which *CYP1B1* gene expression level was also upregulated in T2DM, and whose gene polymorphisms are considered as responsible for increasing T2DM risk [73].

The human apolipoproteins C1 (*APOC1*) and C2 (*APOC2*) are the protein variants of Apolipoprotein E (APOE) constituents of chylomicrons, very low-density lipoprotein (VLDL), and high-density lipoprotein (HDL). *APOC1* is expressed in adipose tissues, brain, liver, spleen, and lungs, and *APOC2* is an important cofactor for lipoprotein lipase activation. Lipoprotein lipase is an enzyme that helps to hydrolyze triglyceride in blood plasma and aids in transferring the fatty acids to all tissues [74,75]. In our pilot study, we observed that *APOC1* and *APOC2* expression levels are highly upregulated in cases compared to controls (Figure 5). This is largely consistent with a previous study in which *APOC1* showed increased expression levels in T2DM [76,77]. Interestingly, *APOC2* has been found to be associated with Alzheimer’s disease [78]. There are proposed models of glucose-mediated neurodegeneration, mostly from studies in Alzheimer’s disease (AD), that provide a mechanistic understanding of the connections between the two diseases [79]. In our study, we speculate that progression to AD could be influenced by insulin signaling pathway (Figure 2), which in turn might lead to the impairment of several crucial cascades, such as synaptogenesis, neurotrophy, and apoptosis, which are regulated by insulin, cholesterol, and glucose metabolism [80], all of which were identified in our results (Figure 2; Table 1).

*MYC,* a proto-oncogene, is known as a transcription factor that helps to promote apoptosis, proliferation, and cell cycle progression. The *MYC* signaling pathways and networks control the utilization of glucose in cells [81]. Previous studies have shown that *MYC* is over-expressed in T2DM and concluded that *MYC* in beta cells activates the apoptosis mechanism, resulting in extensive beta cell loss that ultimately triggers the development of T2DM [82,83]. The *MYC* gene expression is mostly downregulated in our study subjects. Our results are opposite to those of some previous studies that observed that *MYC* gene expression is highly upregulated in diabetes [84,85,86].

Finally, the solute carrier family 2 (*SLC2*) genes are comprised of 14 members, which are essential for the maintenance of glucose uptake and survival of tumor cells. *SLC2A13* and *SLC33A1* are the specific genes of interest from that family based on the IPA knowledge base. There is some information implicating them as potential prognostic biomarkers in acute myeloid leukemia [87]. Both genes were upregulated in all our study subjects (Table 3) with a clear connection to the *Chronic Myeloid Leukemia Signaling* pathway (Figure 2).

### 4.2. Altered Pathways in T2DM

Insulin resistance is the hallmark of diabetes at the cellular level and negatively affects mitochondrial function [88]. Our IPA analysis suggests that several relevant canonical pathways are affected by T2DM, i.e., *Mitochondrial Dysfunctions* and *Insulin Receptor Signaling* (Figure 2). These two pathways may, in turn, trigger additional dysfunctions in intersecting pathways such as *Oxidative Phosphorylation*, *LAX/RXR Activation*, *Sirtuin Signaling*, *Dopamine Receptor Signaling*, *Neuroinflammation Signaling*, and *Gaba Receptor Signaling* (Figure 3), as previously reported in other studies [89].

Our gene expression and IPA analysis showed some evidence that T2DM may affect molecular pathways of relevance to developing cancer. However, the population does not yet show any symptoms of the disease, and hence we view it as a potential future disease risk. The clearest evidence was our unexpected result showing that the tumor suppressor gene *TP53* was affected. *TP53* (tumor suppressor gene) is a transcription factor that mediates the cell’s response to certain stresses and regulates the genes involved in apoptosis, DNA repair, cellular damage, and cell cycle arrest. In our study, this gene was downregulated in nearly all the patients with T2DM, as has been noted in some prior studies [90,91]. *TP53* is also involved in glucose regulation, mitochondrial integrity, fatty acid metabolism, and antioxidant mechanisms [92]. It also regulates the expression of genes including glucose transporter genes Glut 1 and Glut 4 as well as the genes involved in cellular mechanism and glucose homeostasis [93].

The IPA core analysis of significant canonical pathways in T2DM subjects has also highlighted certain interconnected signaling pathways including *P53 Signaling* and *Type 2 Diabetes Insulin Signaling* in tandem with clear manifestations of *Apoptosis Signaling*, *Molecular Mechanism of Cancer*, *Chronic Myeloid Leukemia*, and *Cell Cycle G1/S Checkpoint Regulation. TP53* is a central molecule that connects many of these pathways (Figure 2),s consistent with previous studies [94], and has been associated with increased cancer risks among persons with T2DM [95,96,97,98,99,100].

### 4.3. Strengths and Weaknesses

This study has some strengths, notably, being among the first in Pakistan to investigate altered gene expression in T2DM versus non-diabetic subjects, where we mapped those genes to their affected pathways. It builds upon our prior establishment of a cohort of T2DM patients from different regions of Pakistan [1]. Obesity is one of the major drivers in developing T2DM and is becoming increasingly prevalent in the developing world, including southeast Asian countries like Pakistan and India. According to recent consensus and American Diabetes Association position statements, the BMI cutoff for overweight (≥23 kg/m^2^) and obesity (≥25 kg/m^2^) is lower in Asians [101]. A BMI of 23 kg/m^2^ is then classified as overweight, and a substantial proportion of our T2DM cases therefore fit the overweight and obese categories. We also note that the gold standard for assessing insulin resistance in humans is the hyperinsulinemic-euglycemic clamp. Insulin resistance is generally associated with overweight and obesity, but without having insulin glucose clamp studies at this point, we could not definitively conclude that the participants in our study are insulin resistant. Although the gene expression work was limited by the small sample size, we used a rigorous validation process to highlight the most prominent associations. In the future, larger sample size studies for T2DM are required to confirm these associations and further identify both the underlying mechanisms of disease onset and progression and identifying avenues for treatment and prevention based on such knowledge.

## 5. Conclusions

In conclusion, this small pilot study generated information on potential genetic biomarkers for T2DM through microarray analysis and validated by qRT-PCR TLDA. These genetic variants along with their signaling pathways provide potential insights into certain biological mechanisms and pathogenesis of T2DM in a Pakistani population, which faces a major public health threat from the rising incidence and prevalence of the disease. In the future, larger population-based study designs deploying molecular biomarkers should spur the development of new tools to bridge the current knowledge gaps in understanding how T2DM develops and progresses, and how its impacts on the population can be blunted and ultimately prevented.

## Figures and Tables

**Figure 1 ijerph-17-05866-f001:**
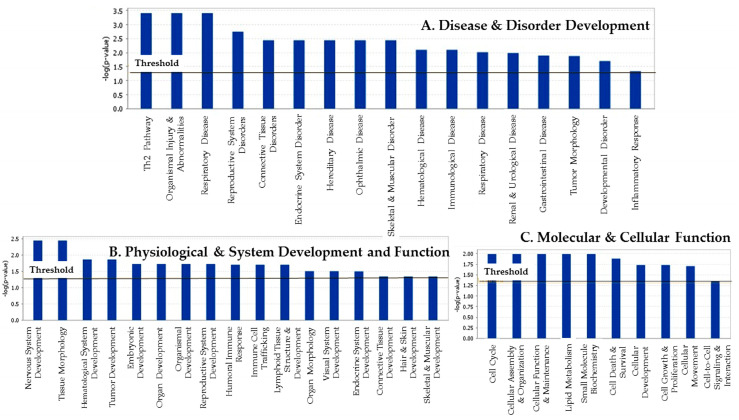
The key bio-functions associated with Type 2 Diabetes subjects including: disease and disorder development (**A**), physiological system development and functions (**B**), and molecular and cellular functions (**C**). The most statistically significant biofunctions that were identified in the IPA core analysis are listed here according to their *p* value (-Log). The threshold line corresponds to a *p* value of 0.05.

**Figure 2 ijerph-17-05866-f002:**
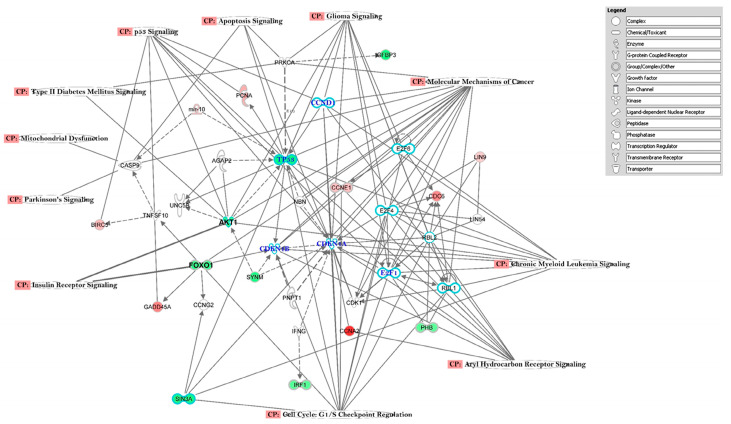
Connectivity of differentially expressed genes in the important signaling pathways in the Type 2 Diabetes subjects, relative to non-diabetic controls, depicting the connectivity between differentially expressed genes (those with ≥ 2-fold change, t-test, *p* < 0.05). Geometric figures in red denote upregulated genes, and those that are green indicate downregulation. Genes in the top 3 networks (from our experimental set of 1657 genes) were allowed to grow our pathway with the direct/indirect relationship from the IPA knowledge base. Solid interconnecting lines show the genes that are directly connected, and the dotted lines signify the indirect connections between the genes and cellular functions. Canonical pathways for signaling that are highly represented are shown within the box. Genes in uncolored notes were integrated into computational generated networks based on evidence stored in the *IPA* knowledge base.

**Figure 3 ijerph-17-05866-f003:**
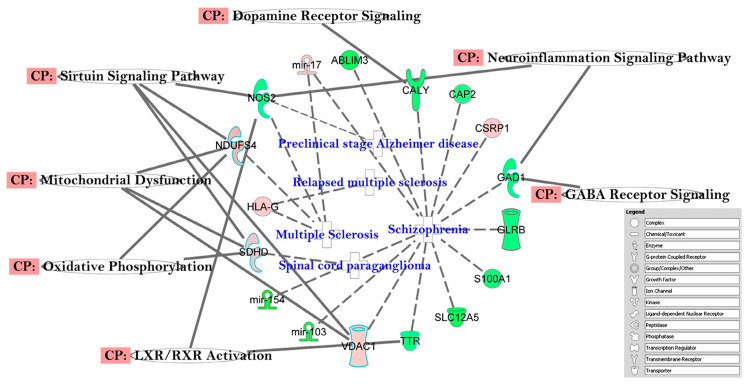
Gene networks created by IPA Analysis that correspond to important canonical pathways reflective of potential future neurological disease/disorders are shown here for our pilot study of subjects with Type 2 Diabetes. Indirect relationships in this context are indicated in blue in the center. Canonical pathways are shown within the box.

**Figure 4 ijerph-17-05866-f004:**
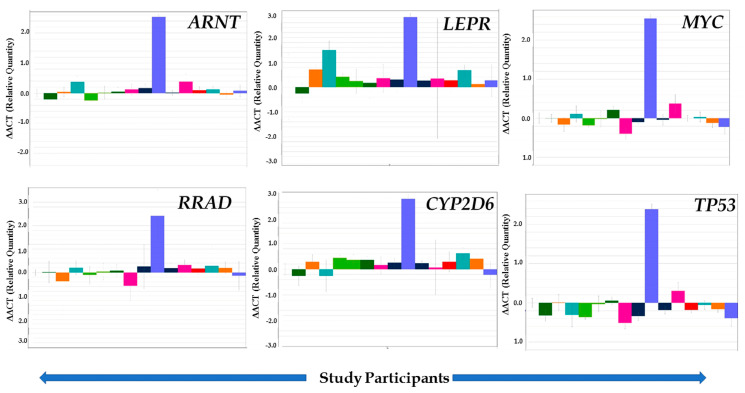
Quantitative real-time PCR (qRT-PCR) results are shown here from our validation of 6 selected genes: *ARNT, LEPR, MYC, RRAD, CYP2D6, and TP53*. Results were obtained by Taqman Low Density Array (TLDA) on the ABI platform (7900HT Fast Real-Time PCR System) after analysis by SDS RQ Manager Version 1.2.1 (^ΔΔ^Ct). Results represented here are the direct snapshots where each panel shows the relative quantification of the selected genes (up- or downregulation) among the study subjects (n = 15 cases with type 2 diabetes and respective controls).

**Figure 5 ijerph-17-05866-f005:**
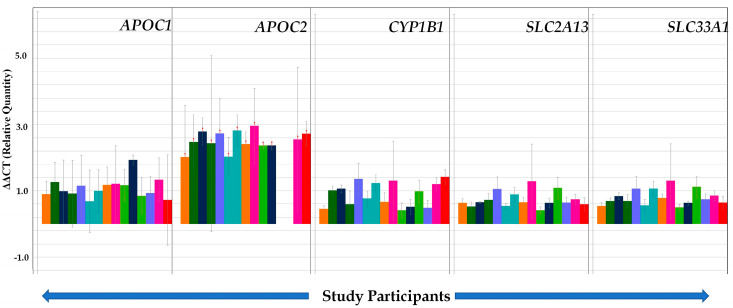
Quantitative real-time PCR (qRT-PCR) results are shown here for validation of *APOC1, APOC2, CYP1B1, SLC2A13, and SLC33A1.* The same methods described under Figure 4 were used to generate these results.

**Figure 6 ijerph-17-05866-f006:**
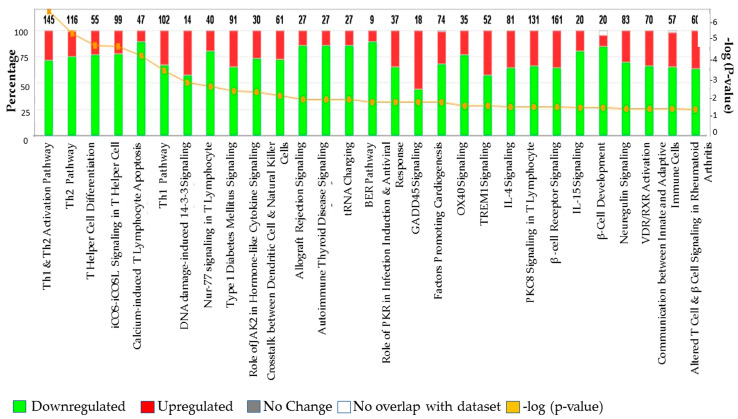
Summary of total upregulated (**red**) and downregulated (**green**) genes representing 29 important signaling and disease pathways in Type 2 diabetes cases compared to non-diabetic controls in the core analysis, reflecting the differential gene expressions obtained from the microarrays (1657 gene sets with ≥ 2-fold change, *t*-test, *p* < 0.05).

**Table 1 ijerph-17-05866-t001:** Summary of the Ingenuity Pathway Analysis (IPA) gene expression analysis results associated with differentially expressed genes in a Pakistani Population with Type 2 Diabetes.

Category	Top Functions and Disease	Significance (*p*-Value)
**Top Canonical Pathways**		
*B-Cell Development/Receptor Signaling*	Immunological Disease/Cell Morphology/Immune Response	8.17 × 10^−6^
*Altered B Cell Signaling*	Hematological System development and Functions	1.70 × 10^−4^
*Allograft Rejection Signaling*	Cell to Cell Communication/Cellular Growth and Proliferation	1.08 × 10^−3^
*Autoimmune Thyroid Disease Signaling*		1.08 × 10^−3^
*Cyclins and Cell Cycle Regulations*	Post-Translational Modification/Cell Cycle/ Connective Tissue Development	1.77 × 10^2^
*Role of NFAT in Regulation of Immune*	Cellular Development, Growth and Proliferation	1.86 × 10^−2^
*Respons* *e*		
*PI3K/AKT Signaling*	Cardiovascular Disease/Cardiovascular System Development and Function	1.99 × 10^−2^
*NK-kB Activation*	Cellular Development, Growth and Proliferation/ Hematological System Development and Function	2.47 × 10^−2^
*PPAR Signaling*	Gene Expression/Cardiovascular System Development and Function	3.24 × 10^−2^
**Important Molecules**		
**Disease and Disorders**		
*Cancer*	**PPAR2**, *TP53*, **UBC**, **AKT1**, **CCL2**, **LRP6**	3.87 × 10^−4^
*Organismal Injury and Abnormality*	**CCL2**, **CD48**, **PARP1**, *CCND2*, **TFP1**, *TP53, YUGB1*	3.87 × 10^−4^
*Respiratory Disease*	**CCL2**, *CUBN, AKT1*, **KIF20A***, CCR6, IGF20A*	3.87 × 10^−4^
*Reproductive System Disease*	*ATP78,***BIRC5**, **CCL2**, **CITED2**, **PSMA1**, *RELA*, TP53	1.97 × 10^−3^
*Connective Tissue Disorder*	*TERT, TP53*	3.59 × 10^3^
*Endocrine System Disorder*	**APCS**, CALR, **EZH2**, PRDX6, TERT, TP53	3.59 × 10^−3^
*Ophthalmic Disease*	*TERT*, *TP53*	3.59 × 10^−3^
*Immunological Disease*	*CD79A, CD79B,***EZH2, LRP6**, *TNFRSF14, TP53*	7.90 × 10^−3^
*Tumor Morphology*	**BIRC5** *, CD40, CD40LG, TP53*	1.31 × 10^−2^
**Molecular and Cellular Functions**		
*Cell Cycle*	**AURKA** *, TERT*	1.03 × 10^−2^
*Cellular Assembly and Organization*	**COL17A1** *, DST*	1.03 × 10^−2^
*Lipid Metabolism*	*ALG2, CD40LG*	1.03 × 10^−2^
*Small Molecule Biochemistry*	*ALG2, CD40LG*	1.03 × 10^−2^
*Cell Death and Survival*	**BIRC5**, *TP53*, **APOD**, *AKT1, SMARCA4*, **COL17A1**	1.31 × 10^−2^
**Physiological System Development and Function**		
*Nervous System Development and Function*	*CNTF,***mir-196**, *OPN1MW*	3.59 × 10^−3^
*Tissue Morphology*	*CNTF, TP53*	3.59 × 10^−3^
*Hematological System Development and Function*	**CCL2**, *FTL3LG, CD4, CD40*	1.31 × 10^−2^
*Tissue Development*	**CCL2**, **DCM1**, *BMP15, ACVR1, ITGA5*	1.37 × 10^−2^
*Endocrine System Development and Function*	**CCNE1**, *TCIM*	3.18 × 10^−2^

Genes in Bold *=* Upregulated; *Italicized =* Downregulated; *p*-value = Fischer’s exact test was used to calculate a *p*-value determining the probability that each biological function and/or disease was different in cases and controls.

**Table 2 ijerph-17-05866-t002:** High-scoring networks identified by Ingenuity^®^ Pathway Analysis in a population of patients with Type 2 Diabetes in Pakistan: Top Five affected networks.

Network ID	Genes in Network	Score	Focus Molecules	Functions
1	**ADH1A, ADH4, APCS, CCNA2, CCNE1, *CD40LG* CDCA2, CDK2**, CDK6, *CDKN18*, **CENPE**, ***CFH*****CKS2**, *CRP*, **FOXMI**, *HNF1A*, ***IL6***, **KIF20A**, ***MIF4GD* PCK1**, PCNA, *TACR3*, ***TGFB1***, *TMED10, AKT1*	14	16	Cell Cycle, DNA Replication, Recombination and Repair, Organismal Survival
2	*ACTL6A, CLAR*, *CAV1*, **CITED2**, *COL1A1*, COL1A2, *FDXR, FUT1*, *HDAC1*, **HUWE1**, **JARID2**, **ENSA**, **EZH2 MCL1**, *MECP2*, **mir-196**, *NOTCH1*, *NTF4*, ***PARP1***, **PRPF8 PIGER3**, *RCOR1*, **SERP1NB2**, *SF1*, *SIN3A*, *SMURF2*, *SNAI1*, *SP1*, *SSPN*, **TBX1**, **TP53**, **TP63**, **TRIM6**, *USP48*, **DNMT1**	10	16	Cellular Development, Organismal Development, Embryonic Development
3	**ADAMTS3**, **ASPM**, **BIRC5**, **CCNB**, **CD44**, DLGAP5 FOXM1, ***FOX01***, **HMMR**, *JAG1*, *JAK1*, *JUN*, **KIF18A**, **KLK6**, **LRP6**, MAPK14, *MITF*, ***MYC***, NCAPG, OSM, ***PBX1***, *PDX1*, *PGR*, *RBPJ*, SERPINA3, ***SLC2A2***, **SPC25**, *TCF7L2*, **TFP1**, **TGFB1**, *TNC*, *TSPAN8*, *TXNIP*, VEGA, *EHF*	10	16	Cancer, Cellular Growth and Proliferation Cellular Development
4	*BID*, **CCL28**, ***CCND2***, **CPT1A**, **CRCP**, *DSG2*, **HCAR3**, **ICAM1**, *IKBKE*, **IL18**, *ITGAV*, *JUN*, *MMD*, **MORF4L1**, *NAUK1*, *OAS1*, *PDPN*, **RRM2**, **TAP1**, ***TGFB*1**, ***TGFBR2***, ***TNF***, *VCAM1*, *VCL*, *WISP1*, *ZNF365*	9	13	Cell-To-Cell Signaling and Interaction, Hematological System Development and Function Inflammatory Response
5	*ACTN4*, ***ATF3***, **CCL2**, ***CD40***, ***CD80***, **CDC6**, CDK6, *CDKN2A*, **COL17A1**, *CXCL8*, *CXCL9*, *DST*, *E2F1*, **ESR2**, *FOS*, *HDAC3*, ***IL13***, ***1L15***, ***IL37***, ***IL6***, ITGA6, **ITGB4**, mir-181, *NCOR2*, **NFKB1**, **NFKB1A**, *RAC1*, RBI, RELA, **SOD2**, *SP1 TERT*, *TLR9*, *TNSFL2*,	8	14	Cancer, Cell Death and Survival Hematological Disease

The genes found to be differentially expressed in our experiments (comparing cases and controls) and the number of such genes displayed in the ‘‘Focus Molecules’’ column have been highlighted in bold print and meet the criteria cutoff and/or filter criteria (IPA Core Analysis) and were mapped along with their corresponding genes derived from IPA Knowledge base (Normal = Upregulated; *Italicized* = Downregulated). The score is generated using a *p*-value (< 0.05). This score indicates the likelihood that the assembly of a set of focus genes in a network could be explained by random chance alone. The data base attributed general cellular functions to each network, which are determined by interrogating the Ingenuity Pathway Knowledge base for relationships between the genes in the network and the cellular functions they impact.

**Table 3 ijerph-17-05866-t003:** Differential expression of genes of interest through relative quantification (^ΔΔ^Ct) that were selected for high-throughput TaqMan Low Density Array (TLDA) card design and their corresponding Probe sets in a small validation study of patients (n = 15) with Type 2 diabetes in Pakistan, relative to corresponding matched controls.

Gene Name (*Probe Sets*)	Descriptions/Functions	Gene Regulation	% Change in Studied Subjects * (Number)	Average Relative Quantification **
***Metabolic Disease and Disorder***				
**LEPR** (*Hs00174492_m1*)	Leptin receptor (Obesity)	Down	7% (n = 1)	−0.35
		Up	93% (n = 14)	+0.66
**RRAD** (*Hs00188163_m1*)	Ras-related associated with diabetes	Down	29% (n = 4)	−0.36+ 0.66
		Up	71% (n = 11)	
**ARNT** (*Hs01121918_m1*)	Encodes a protein that binds to ligand-bound aryl hydrocarbon receptor, involved in xenobiotic metabolism	Down	20% (n = 3)	−0.25
		Up	80% (n = 12)	+0.44
***Neurobehavioral***				
**CYP2D6** (*Hs02576168_m1*)	A member of Cytochrome P450 superfamily enzyme	Down	20% (n = 3)	−0.27
		Up	80% (n = 12)	+0.52
**APOC1** (*Hs03037377_m1*)	Apolipoprotein C1 Family; plays central role in HDL and VLDL metabolism	Up	100% (n = 15)	+1.13
**APOC2** *(Hs000173442_m1)*	Apolipoprotein C2 family that encodes a lipid-binding protein belonging to the apolipoprotein gene family, dysfunction or mutation results into hyperlipoproteinemia type IB, characterized by hypertriglyceridemia, xanthomas, and increased risk of pancreatitis and early atherosclerosis.	Up	87% (n = 13)	+2.54
		*ND*	13% (n = 2)	_
**CYP1B1** *(Hs00164385_m1)*	Cytochrome P450 family 1 subfamily B member 1, which catalyzes many reactions involved in drug metabolism and synthesis of cholesterol, steroids, and other lipids.	Up	100% (n = 15)	+1.12
**SLC2A13** *(Hs00369423_m1)*	Solute carrier family 2 member 13, a member of mitochondrial carrier family	Up	100% (n = 15)	+0.71
**SLC33A1***(Hs00270469_m1*)	Solute carrier family 33 member 1, required for the formation of O-acetylated (Ac) Gangliosides, disorder characterized by congenital cataracts, severe psychomotor retardation, and hearing loss	Up	100% (n = 15)	+0.74
***Cancer***				
**MYC** (*Hs00153408_m1*)	Proto-oncogene, cell cycle progression, apoptosis.	Down	47% (n = 7)	−0.11
		Up	47% (n = 7)	+0.54
		*ND*	6% (n = 1)	-
**TP53** (*Hs01034249_m1*)	Tumor suppressor protein p53	Down	73% (n = 11)	−0.27
		Up	27% (n = 4)	+0.84

****** Data represented as ^ΔΔ^Ct changes (relative quantification) with downregulation (−)/upregulation (+). *Number (n) in parenthesis is the total number of subjects where such changes were observed. % calculation (in parenthesis) was made only among the subjects with amplification under this validation platform. *18s* (*Hs99999901_s1;* manufacturing control), and *GAPDH* (*Hs99999905_m1;* Internal control). *ND*—Not Detected.

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
