# Peer review of "Transcriptional Profiling and Biological Pathway(s) Analysis of Type 2 Diabetes Mellitus in a Pakistani Populationâ€"

_ijerph, 2020, doi:10.3390/ijerph17165866_

Round 1
Reviewer 1 Report
Language in the introduction can be improved. For example the word "exploding" is used incorrectly and the are a lot of extrapolated claims that we cannot be certain will occur. For example on line 64, "The number of people with diabetes will increase by 25% in 2030 and 51% in 2045 [20]." That is a overstated claim. Data trends may purport those numbers but it is not a claim that should be stated with absolute certainty.
In addition much of the introduction is disjoint. There are mentions of.... "environmental risk factors like sedentary lifestyle, physical inactivity [16], high caloric intake, obesity, high consumption of red meat [17], alcohol consumption, smoking, have been recognized as significant contributors" but it is not clear how this ties in with your study or how it is being addressed. If these are contributing factors of T2DM wouldn't it be important to quantify them and explain if and why they might be different in your cases and controls. Further, there is basically zero description of how cases and controls were matched which is extremely important.
More detail also needs to be provided on statistical power and sample size. The article below provides some useful guidance.
Zhao, S., Li, C., Guo, Y. et al. RnaSeqSampleSize: real data based sample size estimation for RNA sequencing. BMC Bioinformatics 19, 191 (2018). https://doi.org/10.1186/s12859-018-2191-5
There are multiple points of confusion from the results that need clarification. First off in Figure 1, Cancer is one of the highest thresholds with a log value of 3.5. You mention..."However, the population does not show any symptomatic outcome for the disease"... Do any of these cases have cancer and are there biomarkers that corroborate with these odds (other than the RNA expression)? If you do not have this data how can you claim this as valid?
A figure showing the physiological pathways for Table 3 would be very helpful. Right now the paper reads almost as if random genes found to be previously related to diabetes were examined and these were tested to see what would stick. I think it would also be beneficial to include additional phenotypical data (e.g. BMI, A1C) that should corroborate with these genetic pathways.
The order of Figures is out of place and hard to follow, the reader has to jump through multiple sections to determine what is being described.
At another point it is stated..."Our validation set of cases and controls revealed that the genes ARNT, LEPR, RRAD and 242 CYP2D6 were strongly upregulated in T2DM subjects compared to controls, except in a few individuals (Table 3; Figure 5, Panels A-D). Following this table I am seeing n of 3-4 which is is almost 25% of the entire sample of 15. Also it would be interesting to know if it is the same 3-4 individuals.
Author Response
Point-by-Point Clarification Note about the Changes in Response to REVIEWER-1
Manuscript ID # ijerph-862967 (Noreen et al.)
Title: Transcriptional Profiling and Biological Pathway(s) Analysis of Type 2 Diabetes Mellitus in Pakistani Population
Authors’ general responses: Authors are grateful for the comments on our article, which have helped us to further improve our manuscript. A point-by-point clarification of the changes made herein is described below towards each concern of the reviewer’s comments. Furthermore, you can view the textual changes in color in the revised manuscript.
Reviewers' Specific Comments and Suggestions for Authors
Concerned Issue 1: Language in the introduction can be improved. For example, the word "exploding" is used incorrectly and the are a lot of extrapolated claims that we cannot be certain will occur. For example, on line 64, "The number of people with diabetes will increase by 25% in 2030 and 51% in 2045 [20]." That is an overstated claim. Data trends may purport those numbers, but it is not a claim that should be stated with absolute certainty.
Response & Measures taken during this revision: Thanks to the reviewer in pointing out some overstatements and the language inconsistencies. We have revised the statement regarding the T2DM scenario in the line 65, and we deleted the sentence that was overstated on the line 64 as “The number of people with diabetes will increase by 25% in 2030 and 51% in 2045”
Concerned Issue 2: In addition, much of the introduction is disjoint. There are mentions of.... "environmental risk factors like sedentary lifestyle, physical inactivity [16], high caloric intake, obesity, high consumption of red meat [17], alcohol consumption, smoking, have been recognized as significant contributors" but it is not clear how this ties in with your study or how it is being addressed. If these are contributing factors of T2DM would not it be important to quantify them and explain if and why they might be different in your cases and controls. Further, there is basically zero description of how cases and controls were matched which is extremely important.
Response & Measures taken during this revision: Authors are indebted to the reviewer for a thoughtful observation about the environmental risk factors or social epigenetics that contribute to T2DM. The study population is likely to be influenced by smoking, high consumption of red meat, and obesity, as we have noted and described in Section 2.1. Study population and Selection of Participants.
Concerned Issue 3: More detail also needs to be provided on statistical power and sample size. The article below provides some useful guidance. Zhao, S., Li, C., Guo, Y. et al. RnaSeqSampleSize: real data based sample size estimation for RNA sequencing. BMC Bioinformatics 19, 191 (2018). https://doi.org/10.1186/s12859-018-2191-5
Response & Measures taken during this revision: For performing microarray expression analysis for our study, we choose to go with at least 2 individuals (Biological Replicates, one male and one female each) with 3 technical replicates (for each subjects) as recommended by manufacturer, and the array console service (Affymetrix Array, by Thermofisher Scientific, CA) that passed all the quality control and statistical measures. While selecting the cases and controls, we were very specific in minimizing the variability contributed by other external factors: the selected subjects had low BMI and were nonsmokers who also reported no alcohol consumption. These selection criteria allowed us to zero in on the genes and pathways associated with T2DM. The Section 2.1, mainly the lines 110-113, have been revised to include this information.
Concerned Issue 4: There are multiple points of confusion from the results that need clarification. First off in Figure 1, Cancer is one of the highest thresholds with a log value of 3.5. You mention..."However, the population does not show any symptomatic outcome for the disease"... Do any of these cases have cancer and are there biomarkers that corroborate with these odds (other than the RNA expression)? If you do not have this data how can you claim this as valid?
Response & Measures taken during this revision: The reviewer’s statement is true to the point that when we completed the cohort study and published our epidemiological information in 2018 (Ref # 1), we did not have any reported cancer cases in the cohort. Our gene expression results, however, indicated “cancer” genes as being differentially expressed in T2DM cases compared to controls, which has revealed through the predictive IPA analysis (Figure 1), network connectivity (figure 2, where TP53 stand as a central molecule in P53 Signaling Pathway), as well as small pilot population validation of TP53 gene (Figure 4), where this tumor suppression gene is mostly down-regulated. Combining all the observed result's criterion, there is a strong indication that continued observation and surveillance of this cohort is warranted in regard to potential cancer risk. These points are now added in the discussion after the line 270-271. as well as with most recent and relevant references # [95-100] linking Type 2 diabetes and cancers in the line 413.
Concerned Issue 5: A figure showing the physiological pathways for Table 3 would be very helpful. Right now the paper reads almost as if random genes found to be previously related to diabetes were examined and these were tested to see what would stick. I think it would also be beneficial to include additional phenotypical data (e.g. BMI, A1C) that should corroborate with these genetic pathways.
Response & Measures taken during this revision: Thanks to the reviewer for suggesting that we include the BMI and HbA1C phenotypic data in the revised manuscript. We have included the information in Supplemental Table 2. The data suggest that the high A1C level is of concern, together with the family history of T2DM in this cohort, and requires careful medical management.
Concerned Issue 6: The order of Figures is out of place and hard to follow, the reader has to jump through multiple sections to determine what is being described.
Response & Measures taken during this revision: Yes, the reviewer was correct that the submitted figures were out of order – we apologize for this technical error and we have corrected it in the revisions.
Concerned Issue 7: At another point it is stated..."Our validation set of cases and controls revealed that the genes ARNT, LEPR, RRAD and 242 CYP2D6 were strongly upregulated in T2DM subjects compared to controls, except in a few individuals (Table 3; Figure 5, Panels A-D). Following this table I am seeing n of 3-4 which is is almost 25% of the entire sample of 15. Also it would be interesting to know if it is the same 3-4 individuals.
Response & Measures taken during this revision. We investigated this issue and found a typo in mentioning figure number, which should now be read as “Figure 4” and it only that reflects the actual genes mentioned here (see Page 8, line 244; corrected now). Likewise, same as “Figure 4” in line 249.
We also checked to see if the same 3 to 4 individuals were affected, as suggested by the reviewer: we did not find any correlation between those specific subjects and their background information and disease status criterion.

Reviewer 2 Report
Noreen et al. examines the molecular pathways between type 2 diabetic participants and non-diabetic controls. Following a microarray and ingenuity pathway analysis (IPA), the authors reported a distinct gene expression profile and signalling pathways. Additionally, the authors validated their findings via qRT-PCR using a small cohort of participants (n=15 per group). Furthermore, the authors reported a gene fingerprint that could lead to the development of biomarkers for high risk patients.
Below are detailed criticisms:
- Obesity is a main driver of type 2 diabetes. Were your controls weight matched? Are the observed findings driven by increased adiposity, rather than insulin resistance? Can you report the participant health parameters and further outline the limitations of these?
- Can you please provide a justification for n = 2 adults? Was a power calculation performed? If this study is under powered potential other candidates may be masked.
- Can Figure 4 and 5 be presented as an average of all type 2 diabetic compared non-diabetic controls? That way all data can be normalised to control.
Minor comments:
Figure 6. Not clear. Both axis are squashed. Figure 6 is also presented in front of figures 2-5. Please re-order.
Author Response
Point-by-Point Clarification Note about the Changes in Response to REVIEWER-2
Manuscript ID # ijerph-862967 (Noreen et al.)
Title: Transcriptional Profiling and Biological Pathway(s) Analysis of Type 2 Diabetes Mellitus in Pakistani Population
Reviewers general comments: Noreen et al. examines the molecular pathways between type 2 diabetic participants and non-diabetic controls. Following a microarray and ingenuity pathway analysis (IPA), the authors reported a distinct gene expression profile and signalling pathways. Additionally, the authors validated their findings via qRT-PCR using a small cohort of participants (n=15 per group). Furthermore, the authors reported a gene fingerprint that could lead to the development of biomarkers for high risk patients.
Authors’ general responses: Authors are thankful for the comments on our article which have helped us to further improve our manuscript. A point-by-point clarification of the changes made herein is described below towards each concern of the reviewer’s comments.
Reviewers' Specific Comments and Suggestions for Authors
Concerned Issue 1: Obesity is a main driver of type 2 diabetes. Were your controls weight matched? Are the observed findings driven by increased adiposity, rather than insulin resistance? Can you report the participant health parameters and further outline the limitations of these?
Response & Measures taken during this revision: We fully accord with the reviewer that obesity as one of the major drivers in developing type 2 diabetes. This is very true for the developed world as it is known establishes and more becoming true for the developing world in south east Asian countries like Pakistan, India, etc. Ideally, BMI should capture adiposity. It is interesting that a constant BMI cutoff for defining obesity (in the true sense, ‘‘adiposity’’) could be applied to all ethnic groups if the composition of body compartments (water, muscle, bone) remained constant in all individuals while body fat alone varied. According to a recent consensus and as per American Diabetic Association position the BMI cutoff (≥23) is lower in Asians (Misra A. Ethnic-Specific Criteria for Classification of Body Mass Index: A Perspective for Asian Indians and American Diabetes Association Position Statement. Diabetes Technol Ther. 2015;17(9):667-671. doi:10.1089/dia.2015.0007). A BMI of 23 is then overweight and our population studied would fit the overweight and obese (BMI 22.73 ± 2.59, (see supplemental Table 2) making them at risk for diabetes. Insulin resistance is generally associated with obesity, but without having insulin glucose clamp studies at this point, we could not definitively conclude that they are or not insulin resistant. This statement is also added appropriately in the Section 4.3.
Concerned Issue 2: Can you please provide a justification for n = 2 adults? Was a power calculation performed? If this study is under powered potential other candidates may be masked.
Response & Measures taken during this revision: It is important to emphasize that this investigation is a pilot study. As we responded to the other reviewer, we wish to verify that for performing microarray expression analysis we choose selected 2 individuals from each of the case and control groups (Biological Replicates, one male and one female each) and with 3 technical replicates for each subject as recommended by manufacturer, and the array console service (Affymetrix Array). The experimental results of the replicates passed all the quality control and statistical criteria. While selecting the cases and controls, we sought minimize the variability contributed by other external factors, including obesity, tobacco smoking, and alcohol consumption, which we treated as exclusion criteria. Therefore, we planned to focus on the associations of genes and pathways solely with T2DM. The Section 2.1, mainly the line 110-113 is revised to accommodate the changes as mentioned herein.
Concerned Issue 3: Can Figure 4 and 5 be presented as an average of all type 2 diabetic compared non-diabetic controls? That way all data can be normalised to control.
Response & Measures taken during this revision: The gene expression data represented here is the direct capture from the QRT-PCR platform with the respective software, wherein the gene expression results for each subject is captured and normalized with their respective matched-control subject samples and replicates in each plate run.
Concerned Issue 4: Figure 6. Not clear. Both axis are squashed. Figure 6 is also presented in front of figures 2-5. Please re-order.
Response & Measures taken during this revision: Also, as mentioned in response to other reviewer, we requested a reorder of the figures and table for a smooth flow of the manuscript.
- Regarding Figure 6, which the reviewer had difficulty viewing, we note that the version that the editorial office sent us reads quite clearly. It is possible that the reviewed version was somehow distorted, and we apologize for that problem.
- A we have replaced the two figures (Figure 1 & 6 with cosmetic changes only, to provide more readable fonts) in their same space, for which we are requesting a re-order of figures as follows:
- A we have replaced the two figures (Figure 1 & 6 with cosmetic changes only, to provide more readable fonts) in their same space, for which we are requesting a re-order of figures as follows:
- Figure 1 should come and align with Table 1, and may appear after Line 213, or suitable.
- Figure 2 and 3 should appear in sequence in the section 3.3 after Line 244.
- Figure 4, 5, & 6 should appear in sequence in Section 3.4; after Line 260.
- We also submitted the high-resolution Figures 1 & 6 online for the revised version.

Round 2
Reviewer 1 Report
The manuscript is much improved and feedback was definitely incorporated.